# M-GCN: A Multimodal Graph Convolutional Network to Integrate Functional and Structural Connectomics Data to Predict Multidimensional Phenotypic Characterizations

**Niharika S. D'Souza** [1], **Mary Beth Nebel** [2,3], **Deana Crocetti** [2,3], **Joshua Robinson** [2], **Stewart Mostofsky** [2,3,4], **Archana Venkataraman** [1]

[1] *Dept. of Electrical and Computer Engineering, Johns Hopkins University, Baltimore, USA*

[2] *Kennedy Krieger Institute, Johns Hopkins School of Medicine, Baltimore, USA*

[3] *Dept. of Neurology, Johns Hopkins School of Medicine, Baltimore, USA*

[4] *Dept. of Psychiatry, Johns Hopkins School of Medicine, Baltimore, USA*

## Abstract

We propose a multimodal graph convolutional network (M-GCN) that integrates resting-state fMRI connectivity and diffusion tensor imaging tractography to predict phenotypic measures. Our specialized M-GCN filters act topologically on the functional connectivity matrices, as guided by the subject-wise structural connectomes. The inclusion of structural information also acts as a regularizer and helps extract rich data embeddings that are predictive of clinical outcomes. We validate our framework on 275 healthy individuals from the Human Connectome Project and 57 individuals diagnosed with Autism Spectrum Disorder from an in-house data to predict cognitive measures and behavioral deficits respectively. We demonstrate that the M-GCN outperforms several state-of-the-art baselines in a five-fold cross validated setting and extracts predictive biomarkers from both healthy and autistic populations. Our framework thus provides the representational flexibility to exploit the complementary nature of structure and function and map this information to phenotypic measures in the presence of limited training data.

**Keywords:** Graph Convolutional Networks, Functional Connectomics, Structural Connectomics, Multimodal Integration, Phenotypic Prediction, Autism Spectrum Disorder

## 1. Introduction

Resting-State functional MRI (rs-fMRI) is a stimulus-free acquisition used to track steady-state changes in co-activation (i.e., connectivity) across the brain (Lee et al., 2013). Complementary to this functional connectivity, Diffusion Tensor Imaging (DTI) captures the directional diffusion of water molecules in the brain as a proxy for structural connectivity (Assaf and Pasternak, 2008). There is mounting evidence in the literature that links the functional signaling and structural pathways in the brain (Skudlarski et al., 2008), with several studies suggesting that this functional connectivity may be mediated by either direct or indirect anatomical connections (Fukushima et al., 2018; Atasoy et al., 2016). Consequently, multimodal integration of connectomics data has become an important topic of study, particularly when characterizing neuropsychiatric disorders, such as autism, ADHD, and schizophrenia (Liu et al., 2015). Traditional multimodal analyses of rs-fMRI and DTI data largely focus on group-wise discrimination. Such methods include statistical tests on edge/node biomarkers (Hahn et al., 2013) to distinguish subgroups in AD, data-driven representations to discriminate schizophrenia patients vs controls (Sui et al., 2013), and

Bayesian models to extract differential networks (Venkataraman et al., 2011, 2013, 2016) While highly informative at the group level, these methods do not directly address inter-individual variability, for example continuous measures of behavior or cognition.

The rise of machine learning has prompted a shift in connectomics towards subject-level predictions. This shift has been accelerated by deep learning, which provides unparalleled representational power. The bulk of deep learning methods focus on diagnostic classification. These approaches range from Multi-Layered Perceptrons (Heinsfeld et al., 2018), Deep Belief Networks (Aghdam et al., 2018), to Convolutional Neural Networks (Khosla et al., 2018). Methods to predict finer-grained characteristics (e.g, demographics or behavior) are sparser and largely focus on a single modality. For example, the authors of (Kawahara et al., 2017) introduced a convolutional neural network that mapped DTI connectivity matrices to cognitive and motor measures. The work of (Lin et al., 2016) proposes an artificial neural network for age prediction from structural connectomes. Finally, the work of (D'Souza et al., 2019) takes the alternative approach of combining a generative dictionary learning framework with a predictive artificial neural network to simultaneously map multiple clinical measures. While these methods achieve good empirical performance, they ignore the interplay between structure and function in the brain. To address this gap, the authors of (D'Souza et al., 2019) extend their framework to combine dynamic rs-fMRI correlations with DTI tractography using a structurally-regularized matrix decomposition (D'Souza et al., 2020). While promising, this method does provide explicit control over the extent to which multi-hop (indirect) structural connections mediate functional connectivity.

Graph neural networks are designed to build representations of nodes and edges within graph structured data, and have found applications in a variety of domains where data naturally assumes a network-like organization (Zhou et al., 2018). These architectures have shown great promise for modeling multi-stage interactions between brain regions that also reflect the hierarchy of brain organization. Hence, these techniques have become important tools in brain connectivity research. Examples include: modeling dynamic functional connectivity for groupwise discrimination (Gadgil et al., 2020), diagnosis of neurodevelopmental disorders (Anirudh and Thiagarajan, 2019; Parisot et al., 2018) from rs-fMRI correlation inputs, or structural connectivity modeling for disease classification (Song et al., 2019). However, current approaches do not leverage the complementarity between the structural and functional graphs or examine dimensional measures of behavior beyond diagnostic classification. We propose a multimodal graph convolutional network (M-GCN) to integrate functional and structural connectivity from rs-fMRI and DTI data respectively, and map this information to phenotypic measures. We employ specialized graph convolutional filters based on (Kipf and Welling, 2016; Kawahara et al., 2017) that operate on functional connectivity inputs, as guided by the subject-level structural graph topology. We demonstrate that our framework generalizes to prediction of phenotypic measures on two separate real world datasets and learns to extract predictive brain biomarkers from limited data.

## 2. Multimodal Graph Convolutional Network for Connectomics

Fig. 1 illustrates our graph convolutional framework, which consists of a representation learning module on the connectomics data (Green Box) cascaded with a fully connected ANN for regression (Blue Box). Let $N$ be the number of patients and $P$ be the number of

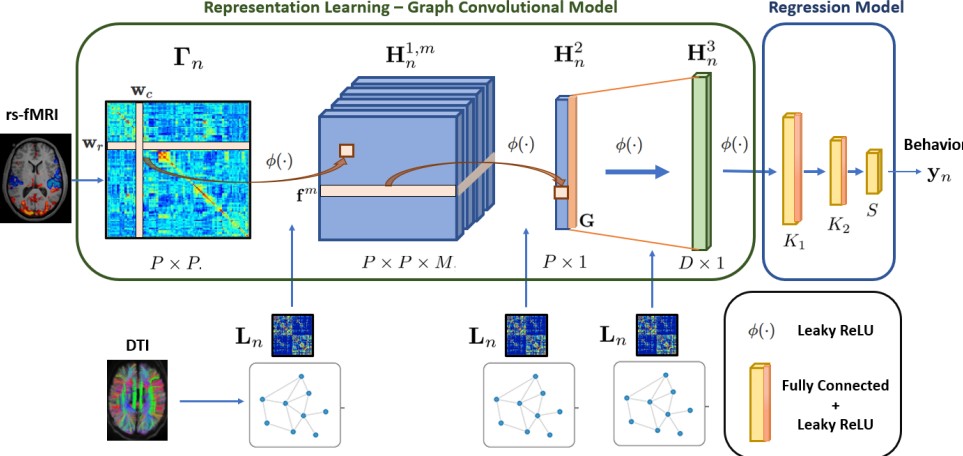

Figure 1: Our M-GCN framework for predicting phenotypic measures **Green Box:** Graph Convolutional Model for Representation Learning from Multimodal Connectomics Data. **Blue Box:** Fully Connected Artificial Neural Network to map to phenotypic measures.

regions in our brain parcellation. Our framework first extracts the structural connectivity graph $\mathcal{G} = (\mathcal{V}, \mathcal{E}_n)$ from DTI tractography. The nodes in $\mathcal{V}$ are brain ROIs defined by the parcellation, while the edges in $\{\mathcal{E}_n\}$ indicate the presence of at least one fiber tract between these regions. Let $\mathbf{A}_n \in \mathcal{R}^{P \times P}$ be the adjacency matrix for $\mathcal{G}$. Correspondingly, we assume that the functional connectivity profile is a signal that rides on the fixed graph montage and is given by rs-fMRI correlation matrices $\mathbf{\Gamma}_n \in \mathcal{R}^{P \times P}$.

Traditional convolutional layers assume a spatial contiguity of the input features, as in the case of 2-D images. This assumption breaks down in general graphs, as node orderings may be arbitrary. Thus, graph convolutional networks define a layer-wise propagation rule designed to aggregate information efficiently at each node based on the underlying graph topology (Bruna et al., 2013; Kipf and Welling, 2016). For a generic input signal $\mathbf{X}^{l-1} \in \mathcal{R}^{P \times C_{l-1}}$, a graph filtering operation can be formulated as follows:

$$\mathbf{X}^l = \phi(\mathbf{L}\mathbf{X}^{l-1}\mathbf{W}) = \phi(\tilde{\mathbf{D}}^{-\frac{1}{2}}\tilde{\mathbf{A}}\tilde{\mathbf{D}}^{-\frac{1}{2}}\mathbf{X}^{l-1}\mathbf{W}) \quad \text{where} \quad \tilde{\mathbf{A}} = \mathcal{I}_P + \mathbf{A}; \ \tilde{\mathbf{D}}_{ii} = \sum_j \tilde{\mathbf{A}}_{ij} \quad (1)$$

where $\mathbf{W} \in \mathcal{R}^{C_{l-1} \times C_l}$ denotes the filter weights, $\mathcal{I}_P$ is an identity matrix of dimension $P$, and $\mathbf{L} = \tilde{\mathbf{D}}^{-\frac{1}{2}}\tilde{\mathbf{A}}\tilde{\mathbf{D}}^{-\frac{1}{2}}$ is the graph Laplacian of the reparameterized adjacency matrix $\tilde{\mathbf{A}}$ and degree matrix $\tilde{\mathbf{D}}$. The authors of (Kipf and Welling, 2016) demonstrate that Eq. (1) is a first order approximation to spectral filtering in the graph Fourier domain.

Inspired by Eq. (1), we define a graph filtering operation that acts on the input functional connectivity matrix $\mathbf{\Gamma}_n$ to generate a connectivity embedding $\mathbf{H}_n^{1,m} \in \mathcal{R}^{P \times P}$ as follows:

$$\mathbf{H}_n^{1,m}(i,j) = \phi\Big((\mathbf{w}_r^m)^T \mathbf{L}_n \mathbf{\Gamma}_n(:,j) + \mathbf{\Gamma}_n(i,:)\mathbf{L}_n \mathbf{w}_c^m + \mathbf{b}^1\Big) \quad m \in \{1, \dots M\} \quad (2)$$

Here, $M$ is the number of channels, each parametrized by a row and column filter $\mathbf{w}_r^m, \mathbf{w}_c^m \in \mathcal{R}^{P \times 1}$ and a bias term $\mathbf{b}^1 \in \mathcal{R}^{P \times 1}$, resulting in a total of $(2P + 1)$ learnable parameters

per channel. Effective, $\mathbf{H}_n^{1,m}(i,j)$ computes a weighted sum of the functional connectivity profile of nodes $i$ and $j$, further regularized by the DTI graph Laplacian $\mathbf{L}_n$. Conceptually, Eq. (2) is similar to the cross shaped E2E filters in (Kawahara et al., 2017). We also note that, despite the symmetry of the correlation matrices $\mathbf{\Gamma}_n$, the embedding $\mathbf{H}_n^{1,m}$ can be assymmetric. This allows us to account for any laterality in functional subsystems.

Following the connectome embedding in Eq. (2), we use two more graph convolutional layers with pooling to first compute a node-wise representation $\mathbf{H}_n^2 \in \mathcal{R}^{P\times 1}$ and a whole-graph embedding $\mathbf{H}_n^3 \in \mathcal{R}^{D\times 1}$. Mathematically, these operations can be represented as:

$$\mathbf{H}_n^2 = \phi\Big(\sum_m \mathbf{L}_n \mathbf{H}_n^{1,m} \mathbf{f}^m + \mathbf{b}^2\Big) \qquad \mathbf{H}_n^3 = \phi\Big(\mathbf{G}\mathbf{L}_n \mathbf{H}_n^2 + \mathbf{b}^3\Big) \tag{3}$$

The filter weights are parameterized by the vectors $\mathbf{f}^m \in \mathcal{R}^{P\times 1}$ per $M$ channel, the graph embedding matrix $\mathbf{G} \in \mathcal{R}^{D\times P}$, and the bias terms $\mathbf{b}^2$ and $\mathbf{b}^3$ respectively. In total, these layers add another $(M+D)P+2$ learnable parameters. Eq. (3) parallels the computation of centrality measures in graph theoretic literature by summarizing node-wise information based on functional similarity, as guided by structure. Finally, our graph embedding $\mathbf{H}_n^3$ is input to an ANN to map to the phenotypic measures $\mathbf{y}_n \in \mathcal{R}^{S\times 1}$ for patient $n$. The ANN is a simple three layered fully connected network of sizes $D \times K_1$, $K_1 \times K_2$ and $K_2 \times S$.

**Implementation Details:** We train our M-GCN on a combination of $\ell_2$ loss and $\ell_1$ loss between the predicted $\hat{\mathbf{y}}_n$ and true measures $\mathbf{y}_n$:

$$\mathcal{L} = \frac{1}{NS} \sum_{n=1}^N \Big[ ||\mathbf{y}_n - \hat{\mathbf{y}}_n||_2 + ||\mathbf{y}_n - \hat{\mathbf{y}}_n||_1 \Big] \tag{4}$$

The $\ell_1$ loss function has been shown to be more robust to outliers as compared to the $\ell_2$ loss (Qi et al., 2020), but less stable during training due to the lack of smoothness near the optimal solution (Friedman et al., 2001). We found that this combined loss empirically provided a good tradeoff between stability and generalization. Layer sizes for the M-GCN were set to $M = 32$ channels for the connectome embedding, $D = 256$ for the graph embedding and $\{K_1, K_2\} = 128, 30$, as we found these choices to be sufficient to map the connectomics data to the phenotypic measures during training. We chose a LeakyReLU ($\phi(x) = \max(0, x) + 0.1 * \min(0, x)$) as the activation function with our network layers, which we found empirically robust to saturation and exploding gradients during training. We train our M-GCN via stochastic gradient descent (SGD) algorithm with momentum ($\delta = 0.9$), batch size $= 16$, with an initial learning rate of 0.001 decayed by 0.9 every 10 epochs. Additionally, we utilize a weight decay of 0.001 as regularization and train our network for 40 epochs to avoid overfitting. All parameters were determined based on a validation set of 30 additional patients from the HCP dataset. We carried forward the same settings to the second KKI dataset.

## 2.1. Baselines

We compare the predictive performance of our network against the following baselines:

**Multimodal ANN:** We use a four layer ANN that maintains the same number of parameters, activation, and loss function as the M-GCN. It operates on the vectorized $P \times (P-1)/2$

rs-fMRI correlations, each multiplied by the corresponding entry of the DTI Laplacian $\mathbf{L}_n$. This baseline evaluates the benefit of maintaining the graph structure of the data.

**rs-fMRI only GCN:** We use the same architecture as our M-GCN but omit the graph Laplacian in Eqs. (2-3). This baseline evaluates the benefit of DTI regularization.

**BrainNetCNN:** We integrate multimodal connectivity data via the BrainNetCNN (Kawahara et al., 2017), originally designed to predict cognitive outcomes from DTI data. We modify this architecture to have two branches, one for the rs-fMRI correlation matrices $\mathbf{\Gamma}_n$, and another for the DTI Laplacians $\mathbf{L}_n$. The ANN is modified to output $S$ measures of clinical severity. We set the hyperparameters according to (Kawahara et al., 2017)

**Dictionary Learning + ANN:** The integrated framework in (D'Souza et al., 2019) uses static rs-fMRI correlation matrices ($\mathbf{\Gamma}_n$) to simultaneously predict multiple clinical or behavioral measures. The model combines a dictionary learning generative term with a neural network predictor. The two blocks are optimized jointly in an end-to-end fashion.

**Dynamic Deep-Generative Hybrid:** The framework in (D'Souza et al., 2020) uses a similar joint optimization strategy but operates on dynamic rs-fMRI correlation matrices $\{\mathbf{\Gamma}_n^t\}$ and incorporates DTI regularizer in the dictionary learning term. Overall, these last two baselines evaluate the benefit of GCNs for implicit representational learning over a classical decomposition strategy. We have followed the guidelines provided by the authors to set the hyperparameters and train both of these baselines.

## 3. Experimental Evaluation and Results

### 3.1. Datasets and Pre-processing

**HCP Dataset:** Our first dataset contains 275 healthy individuals from the Human Connectome Project (HCP) S1200 database (Van Essen et al., 2013). Rs-fMRI and DTI scans are acquired on a Siemens 3T scanner (**rs-fMRI:** EPI, TR/TE= $0.72ms/0.33ms$, flip angle $= 52$, res $= 2$mm$^3$, duration $= 1200$ time samples per run; **DTI**: EPI, SENSE factor $= 1$, TR/TE $= 5520/89.5$ms, res $= 1.25 \times 1.25 \times 1.25$mm, b-value $= 1000/2000/3000s/mm^2$ interleaved, with 95/96/96 gradient directions respectively). To remain commensurate with clinical scanning protocols, we selected a 15-minute interval from the rs-fMRI scans for our analysis. Rs-fMRI data was pre-processed according to the standard HCP pipeline (Smith et al., 2013), which accounts for motion and physiological confounds. DTI data was processed using the standard Neurodata MR Graphs package (Kiar et al., 2016), which uses streamline tractography to estimate fiber bundles. Our phenotypic measure was the Cognitive Fluid Intelligence Score (CFIS) (Duncan, 2005; Bilker et al., 2012) adjusted for age, which is obtained via a battery of tests measuring cognitive reasoning (dynamic range: $70 - 150$)

**KKI Dataset:** Our in-house clinical dataset was acquired at the Kennedy Krieger Institute. It consists of 57 children with high-functioning ASD. Rs-fMRI and DTI scans were acquired on a Philips $3T$ Achieva scanner (**rs-fMRI:** EPI, TR/TE $= 2500/30$ms, flip angle $= 70$, res $= 3.05 \times 3.15 \times 3$mm, duration $= 128$ or $156$ time samples; **DTI**: EPI, SENSE factor $= 2.5$, TR/TE $= 6356/75$ms, res $= 0.8 \times 0.8 \times 2.2$mm, b-value $= 700s/mm^2$, 32 gradient directions). Our rs-fMRI preprocessing includes motion correction, normalization

to the MNI template, spatial and temporal filtering, and nuisance regression with Comp-Corr (Behzadi et al., 2007). We use the FDT pipeline in FSL to pre-process the DTI scans (Jenkinson et al., 2012). Tractography is performed using the BEDPOSTx and PROB-TRACKx functions in FSL (Behrens et al., 2007). We use three separate clinical batteries to characterize various impairments associated with ASD. The Autism Diagnostic Observation Schedule (ADOS) (Payakachat et al., 2012) measures socio-communicative deficits and restricted/repetitive behaviors via a behavioral evaluation (dynamic range: $0 - 30$). The Social Responsiveness Scale (SRS) (Payakachat et al., 2012) quantifies impaired social functioning via a parent/teacher questionnaire (dynamic range: $70 - 200$). Finally, Praxis (Dziuk et al., 2007; Mostofsky et al., 2006) measures the ability to perform skilled motor gestures on command and is scored by two research reliable raters (dynamic range: $0 - 100$).

For both datasets, we use the Automatic Anatomical Labeling (AAL) atlas (Tzourio-Mazoyer et al., 2002) to define 116 cortical, sub-cortical and cerebellar brain ROIs for both the functional and structural connectivity matrices. We also subtract the first eigenvector from the rs-fMRI correlation matrices, which is a roughly constant bias, and use the residual matrices as the inputs to all models.

### 3.2. Performance Characterization.

**Predicting CFIS:**  Table 1 (and Fig. 2-Appendix) illustrates our method and baselines for predicting CFIS for the HCP dataset in a five-fold cross validated setting. We quantify the performance via the Median Absolute Error (MAE), the Normalized Mutual Information (NMI) and the Coefficient of Correlation (R Stat.) between the actual and predicted measures. Lower MAE and higher NMI/R Stat. indicate better performance. The training performance is good for all methods. However, the M-GCN clearly outperforms the baselines when generalizing to unseen testing data. As a benchmark, our validation performance (Test MAE: 13.41 ± 8.17, NMI Test: 0.71, R: 0.42) also provides similar generalization.

**Multidimensional Clinical Severity Prediction:**  Table 2 (and Fig. 2 in the appendix) compares the multi-output prediction performance of ADOS, SRS, and Praxis on the KKI dataset for a five fold cross validation. Again, we observe that the M-GCN outperforms the baselines for the prediction of all three severity measures in almost every case. Note that, from a clinical standpoint generalization to prediction of multiple deficits is inherently

| Meas. | Method | MAE Test | NMI Test | R Stat. | p |
|-------|--------|----------|----------|---------|---|
| CFIS | Mult. ANN | 14.06 ± 10.16 | 0.61 | 0.23 | 0.065 |
| | rs-fMRI only GCN | 14.16 ± 8.96 | 0.54 | 0.23 | 0.044* |
| | BrainNetCNN | 17.90 ± 17.55 | 0.58 | 0.25 | 0.0015* |
| | Dict. Learn. + ANN | 15.26 ± 13.99 | 0.66 | 0.29 | 0.024* |
| | Dyn. Deep-Gen. Hyb. | 16.31 ± 15.43 | 0.67 | 0.30 | 0.0043* |
| | **Our Framework** | **12.87 ± 9.65** | **0.73** | **0.41** | - |

Table 1: **HCP Dataset:** Evaluation using the **Median Absolute Error (MAE)**, **Normalized Mutual Information (NMI)** and **R Statistic** for the test set. Best performance is highlighted in bold. Near misses are underlined p value (**p**) for differences in distribution of the test MAE of the M-GCN against the baselines via the t test. ∗ denotes $p < 0.05$.

| Meas. | Method | MAE Test | NMI Test | R Stat. | p |
|-------|--------|----------|----------|---------|---|
| ADOS | Mutl. ANN | 2.96 ± 2.30 | 0.30 | 0.04 | 0.041* |
| | rs-fMRI only GCN | 3.14 ± 2.25 | 0.41 | 0.16 | 0.002* |
| | BrainNetCNN | 3.50 ± 2.20 | 0.25 | 0.41 | 0.009* |
| | Dict. Learn. + ANN | **2.71 ± 2.40** | 0.43 | **0.50** | 0.20 |
| | Dyn. Deep-Gen. Hyb. | 2.84 ± 2.79 | 0.34 | 0.47 | 0.10 |
| | **Our Framework** | **2.71 ± 2.15** | **0.45** | **0.50** | - |
| SRS | Mult. ANN | 18.47 ± 11.04 | 0.60 | 0.03 | 0.033* |
| | rs-fMRI only GCN | 21.34 ± 8.58 | 0.62 | 0.16 | 0.019* |
| | BrainNetCNN | 18.96 ± 15.65 | 0.75 | 0.13 | 0.039* |
| | Dict. Learn. + ANN | 16.79 ± 13.83 | **0.89** | **0.37** | 0.13 |
| | Dyn. Deep-Gen. Hyb. | 17.81 ± 16.09 | 0.88 | 0.30 | 0.073 |
| | **Our Framework** | **16.50 ± 9.44** | 0.85 | 0.35 | - |
| Praxis | Mult. ANN | 17.12 ± 16.66 | 0.65 | 0.25 | 0.008* |
| | rs-fMRI only GCN | 16.71 ± 16.66 | 0.74 | 0.17 | 0.019* |
| | BrainNetCNN | 15.15 ± 11.49 | 0.19 | 0.3 | 0.024* |
| | Dict. Learn. + ANN | 13.19 ± 10.75 | 0.82 | 0.37 | 0.15 |
| | Dyn. Deep-Gen. Hyb. | 13.50 ± 11.55 | 0.85 | 0.31 | 0.089 |
| | **Our Framework** | **12.82 ± 12.04** | **0.86** | **0.46** | - |

Table 2: **KKI Dataset:** Evaluation using the **Median Absolute Error (MAE)**, **Normalized Mutual Information (NMI)** and **R Statistic** for the test set. Best performance is highlighted in bold. Near misses are underlined p value (**p**) for differences in distribution of the test MAE of the M-GCN against the baselines via the t test. ∗ denotes $p < 0.05$.

more challenging than predicting a single phenotypic measure. This also partially accounts for the poor performance of some of the baselines, where they perform reasonably well for the prediction of one of the measures (for example, the rs-fMRI only GCN for ADOS), but at the expense of generalization onto the other two measures. Overall, our experiments on two different real world datasets allude to reproducibility and suggest that the M-GCN generalizes effectively even with modest training sample sizes. Moreover, the performance gains against the M-GCN baseline without the DTI indicate the benefit provided by the multimodal integration via our graph convolutional framework.

**Extracting Clinical Biomarkers:** The representations learned by the row and column filter pairs $\mathbf{w}_r$ and $\mathbf{w}_c$ at the input layer of the M-GCN (i.e. Eq. (2)) may illuminate key biomarkers for each population. We first match the filter pairs across the cross validation folds based on the average correlation coefficient between the row and column filter weights. Fig. 2 illustrates four filter pairs out of 32 that appear most frequently across subsets of the HCP and KKI dataset. In each case, we plot the average row filter (RF) and column filter (CF) weights projected onto the corresponding regions of the AAL atlas. Compared with the filters learned by the rs-fMRI only GCN (Appendix Fig. 3), the DTI regularization in the M-GCN offers sparsity and better spatial selectivity in the patterns captured.

For the HCP dataset (Fig. 2 (a)), we observe that RF1, RF2, CF1 and CF2 display contributions from regions of the Default Mode Network (DMN), known to play a critical role in consolidating working memory (Sestieri et al., 2011) and is widely inferred within the resting state literature. RF3 and CF3 highlight regions of the Frontoparietal Network (FPN)

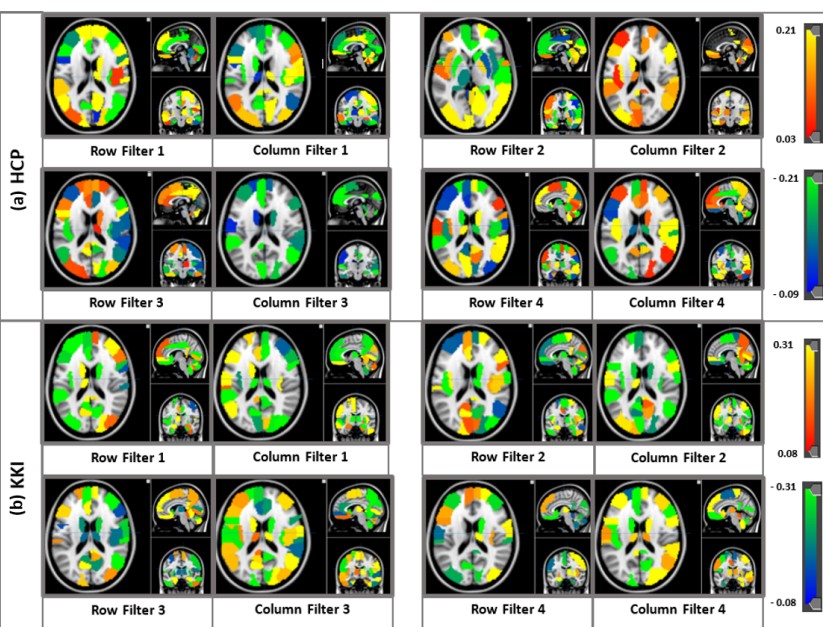

Figure 2: Four pairs of row & column filter weights learned by the M-GCN on the (a) HCP dataset and (b) KKI dataset. The colorbar quantifies the filter weight for each AAL ROI.

and the Medial Prefrontal Network (MPN), believed to play a role in working memory, attention and decision making, which are associated with cognitive intelligence (Menon, 2011). CF4 highlights regions from the Somatomotor Network (SMN) while RF4 includes subcortical and cerebellar regions. Together, these are believed to be important functional biomarkers of cognitive intelligence in literature (Chén et al., 2019). For the KKI dataset (Fig. 2 (b)), we observe that RF1, CF1, CF2 and CF4 highlight areas from the DMN and SMN. Altered connectivity within these regions is widely reported in ASD literature (Nebel et al., 2016). RF3, RF4 and CF4 also highlight contributions from the higher order visual processing areas and sensorimotor regions, which are in line with findings of reduced visual motor integration in Autism (Nebel et al., 2016). RF3, RF4 and CF4 also display contributions from subcortical regions along with the prefrontal cortex and DMN, which is believed to be relevant to social-emotional regulation in ASD (Pouw et al., 2013).

## 4. Conclusion

We have introduced a novel multimodal graph convolutional framework to leverage complementary information from functional and structural connectivity. Our M-GCN is designed to effectively utilize the underlying anatomical pathways to learn rich representations from functional connectivity data that are simultaneously informative of multidimensional phenotypic characterizations. We demonstrate that this framework is able to learn effectively from limited training data and generalize well to unseen patients. Finally, our framework makes minimal assumptions, and can potentially be applied to study other neuro-psychiatric disorders (eg. ADHD, Schizophrenia) as a diagnostic tool.

## Acknowledgments

This work is supported by the National Science Foundation CRCNS award 1822575 and CAREER award 1845430, the National Institute of Mental Health (R01 MH085328-09, R01 MH078160-07, K01 MH109766 and R01 MH106564), the National Institute of Neurological Disorders and Stroke (R01NS048527-08), and the Autism Speaks foundation.

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
