# OpenReview forum: "M-GCN: A Multimodal Graph Convolutional Network to Integrate Functional and Structural Connectomics Data to Predict Multidimensional Phenotypic Characterizations"
_MIDL.io/2021/Conference — MIDL 2021_

### Official Review · AnonReviewer4 · 2021-03-07

**Confidence:** 5
**Preliminary Rating:** 2
**Final Rating:** 3

**Summary:**

This paper presents a multimodal spectral graph filtering approach for combining functional and structural connectivity within a single graph convolutional model. The method is well presented and the results are validated against state-of-the art multimodal ANNs and GCNs for two difficult problems of cognitive test prediction, on a small subset of the HCP and on an Autism.


**Strengths:**

The ideas in this paper are well motivated, and there is no argument that new methods which effectively integrate dMRI and fMRI would be useful for a number of different problems.

The methods are clearly presented. The figures are well made and clearly described in the text. The method is validated against several other methods which would be considered state of the art for this domain. Results do seem improved relative to baselines. However, sometimes the differences are marginal


**Weaknesses:**

My slight concern with this paper is whether it would still outperform other methods if they were to use the HCP data, processed in the most subject specific sensitive way.

Here the authors state that they use only 275 out of ~ 1000 HCP subjects and only 15mins instead of 1hr of resting state fMRI. It is well known that fMRI connectivity matrices improve in stability with more data. They also process data in the volume rather than using the image processing advances of the HCP pipeline, which is designed to improve the group-wise consistency of fMRI data on the surface. Finally why use the AAL atlas rather than the HCP parcellation which is optimised for cortical organisation. While the authors state that this is to put the model on comparable settings to clinical data. Would it not be good to first test how well it works on the best quality data, especially when the goal is to predict such a challenging phenotype as fluid intelligence? It would actually be good to have a graph plotting the effect of performance relative to the data sets size and length of fMRI.

I would be less concerned about this if the results showed stronger improvement relative to past methods however the reported differences are small and it is difficult to tell whether the reported improvements are statistically significant?

Also I think the small data sets used here are a problem because the plots (figure 4 especially) seem to suggest that the model is mostly predicting the mean for the test examples - suggesting overfitting?


**Deanonymize Review:**

no

**Final Rating Justification:**

The methodological contribution is sufficient but the validation could still be more thorough.

**Justification Of The Preliminary Rating:**

I think this paper clearly has merits, the methods are novel and interesting and the experiments validate against several recent competing methods. My concern is that the HCP results seem overfit and the Autism results show marginal improvements relative to other methods. I believe it would be a more convincing paper if they validated on a larger HCP dataset with longer fMRI datasets in order to overcome overfitting and presented significance tests for all results.




**Paper Type:**

methodological development

**Questions To Address In The Rebuttal:**

1. How does the method perform for the full HCP data set - or at the very least can over fitting be reduced with higher numbers and/or more fMRI data
2. Some significance testing on the differences between results for different methods in the paper might help, especially for the Autism table where often the differences seem marginal


**Special Issue:**

no

---

### Official Review · AnonReviewer1 · 2021-03-07

**Confidence:** 5
**Preliminary Rating:** 3
**Recommendation:** Oral, Poster
**Final Rating:** 3

**Summary:**

The authors propose to infer various cognitive and social measures from resting-state fMRI and DTI data with a graph convolutional neural network (GCN). The GCN takes as input a functional connectivity matrix and regularizes each layer with the DTI connectivity matrix. The GCN outputs an embedding which is fed to a fully connected network to predict the corresponding metric.

Two datasets are used for the experiments: 275 subjects from the HCP1200 dataset, as well as an in-house dataset. The proposed method is compared against ablated versions of the proposed network, prior work from the first author and external prior work. Finally, the authors highlight correlation between brain regions and the learned weights of the network according to specific metrics, which are validated by existing literature.

**Strengths:**

This is an abstract of very high quality. It is is clear, concise but does not cut corners. The presented work is well described, the loss function is explicitly stated as are the graph convolutional operations. The inputs and outputs of the network(s) are well presented and the figures are helpful. Each comparison method is well defined and relevant, including the "ablated" versions of the proposed method.  Two datasets were used for the experiments and each experiment is explained as well as they should be. Results are complete, well presented and meaningful.


**Weaknesses:**

It is in the opinion of the reviewer that besides minor comments, the main weakness of the paper is in the presentation of the results. Performance measures at training time for both experiments and all metrics are reported, but only contribute in hiding how the proposed work performs better at test time than the other methods. While transparency is important, test time performance is arguably what is most important, and if the results at train time were to be removed, the proposed work would shine even more.

**Deanonymize Review:**

no

**Detailed Comments:**

## Abstract

> We validate our framework on a dataset of 275 healthy individuals from the Human Connectome Project database to map to cognition and on an independent clinical dataset of 57 individuals diagnosed with Autism Spectrum Disorder
to predict multiple behavioral deficits.

The term "to map to cognition" is a bit convoluted. The sentence could probably be improved by joining the two datasets and experiment description, e.g.

> We validate our framework on 275 healthy individuals from the Human Connectome Project and 57 individuals diagnosed with Autism Spectrum Disorder from an in-house data to predict multiple cognitive and social measures

## Section 1

>  Examples include, modeling dynamic functional [...]

The comma should be replaced by a colon.
## Section 2

In equations 1, 2 and 3, $\phi$ is mentioned but not defined anywhere.

## Section 3.2

In Table 1, in the column NMI Train, the result for "Dict. Learn + ANN" is highlighted. However, "Mult. ANN" reports a higher value (0.93).

The caption of Figure 2 could be improved so that its content is (more) interpretable on its own. For example, by labelling the values on the right.

**Final Rating Justification:**

The changes made by the authors have greatly improved the quality of the presented work by adding more structure and rigor to the text.

**Justification Of The Preliminary Rating:**

While some comments were made on specific aspects of the abstract, it represents nonetheless work of high quality, both in the proposed work as well as the abstract and I believe the abstract is very well suited to be presented at MIDL 2021.

**Paper Type:**

both

**Questions To Address In The Rebuttal:**

Points brought forward in the "Weaknesses" section as well as the "Detailed comments" should be addressed.

**Special Issue:**

no

---

### Official Review · AnonReviewer2 · 2021-03-08

**Confidence:** 4
**Preliminary Rating:** 4
**Recommendation:** Oral, Poster

**Summary:**

This paper presents an extension of the authors' GCN work that combines functional and diffusion connectome information applied to a task of predicting symptoms in an ASD study. The proposed multi-modal GCN ( called M-GCN) act topologically on the functional connectivity matrices and is guided by the subject-wise structural connectomes.  The evaluation is rather broad with a set of alternative methods. Overall the addition of the structural connectome to the MGCN provides significant improvement in the results.

**Strengths:**

- interesting topic, nice GCN task
- clean presentation
- nice evaluation with multiple alternative methods
- novel (albeit iterative) method (Multimodal ANN, rs-fMRI only GCN, BrainNetCNN, Dictionary Learning + ANN, Dynamic Deep-Generative Hybrid GCN), with the proposed method showing improved results in most comparisons.
- interesting results on prediction single vs multiple continuous symptom scores

**Weaknesses:**

The proposed research is somewhat iterative over existing graph convolutional networks, including the authors' prior work.
- unclear about which comparative results reach statistical differences among the methods


**Deanonymize Review:**

no

**Justification Of The Preliminary Rating:**

Overall very nice research with a good application throughout, though with a bit of iterative novelty. The evaluation is  nice and quite broad for a conference paper. Overall I would be interested to see this paper presented at the conference.

**Paper Type:**

methodological development

**Special Issue:**

yes

---

### Official Review · AnonReviewer3 · 2021-03-10

**Confidence:** 3
**Preliminary Rating:** 3
**Final Rating:** 3

**Summary:**

Authors propose a multimodal graph convolutional network to predict phenotypic measures. The main novelty is a graph filtering operation that integrates resting state fMRI connectivity and diffusion connectivity. Authors demonstrate the effectiveness of the proposed method on two different datasets.

**Strengths:**

-	The proposed method is interesting and well explained
-	Authors carefully present enough mathematical and algorithmic details to make the proposed method highly reproducible
-	Authors present both quantitative and qualitative results

**Weaknesses:**

-	Some technical terms are not sufficiently explained
-	Important differences between train and test metrics seem to indicate overfitting. Authors should better discuss this point.
-	More details about the computation of the edges of the diffusion connectivity matrix are needed

**Deanonymize Review:**

no

**Final Rating Justification:**

Authors clarified all my concerns and I confirm my positive score.
However, I invite authors to add this point in the paper:

Quantitative Performance: To restate our evaluation approach, we first set hyperparameters for the M-GCN (learning rate schedule, number of epochs etc) via a separate validation set from the HCP database. Then, using these settings, we train and test our models separately on the HCP and KKI datasets in a five-fold cross validation setting, and report the performance in Tables 1 and 2 respectively. If we observe the final performance on the HCP validation set (Val. MAE: 13.41 +/- 8.17, Val. NMI: 0.71 Val. R. :0.42) against the HCP test metrics in Table 1 (Test MAE: 12.87 +/- 9.65, Test NMI: 0.73, Test R: 0.41), we notice that these metrics are pretty comparable. This suggests that the test performances in Tables 1 and 2 are reflective of generalization onto unseen data.

**Justification Of The Preliminary Rating:**

The paper is well written and interesting but some points need to be better explained or discussed.

-	What do the authors mean with multi-stage structural pathways and interactions?
-	Do the authors use the same graph (i.e. vertex definition, parcellation) for both functional and connectivity matrix? I guess so, but authors should probably consider better explaining this point.
-	Why did the authors use binary edges instead than scalar ones? If I correctly understood, the weight of an edge is equal to 0 if there is no fiber connecting two vertices, otherwise it’s 1. Why not using the number of fibers connecting two regions as weight for an edge? Authors should better discuss this point.
-	Authors use a kind of elastic net loss in Eq.4. Do the authors use a hyper-parameter (weight) between the two terms?
-	If I correctly understood, authors show in Table 1 and Table 2 the results of two different models learnt on the HCP and KKI datasets respectively. In both cases, the results between training and test are very different. Is this due to a large overfitting? Limited sample size? Authors should better discuss this point.


**Paper Type:**

methodological development

**Special Issue:**

no

---

### Meta-Review · Area_Chair1 · 2021-03-28

**Recommendation:** Accept (Oral & Special Issue Candidate)

**Metareview:**

The paper proposes a graph convolutional network that simultaneously operates on two interdependent networks, namely structural and functional brain connectivity networks. This is both relevant and challenging. The paper is well written and presents supporting experiments. There is some criticism of the experiments for not using the specialized HCP preprocessing, but rather approaching a more typical clinical preprocessing scheme, leaving questions of whether the method would still outperform state of the art if the network had the data quality implied by the HCP preprocessing. The results are, nevertheless, relevant, as most clinical (or even research) data also does not have HCP quality.

**Paper Type:**

methodological development

---

### Decision · Program_Chairs · 2021-03-31

**Decision:**

Accept

**Comment:**

Congratulations your paper has been selected as a long oral.